# SPTK4: An Open-Source Software Toolkit for Speech Signal Processing

*Takenori Yoshimura[1], Takato Fujimoto[1], Keiichiro Oura[2], and Keiichi Tokuda[1]*

[1]Nagoya Institute of Technology, Nagoya, Japan
[2]Techno-Speech, Inc., Nagoya, Japan

yoshimura.takenori@nitech.ac.jp

## Abstract

The Speech Signal Processing ToolKit (SPTK) is an open-source suite of speech signal processing tools, which has been developed and maintained by the SPTK working group and has widely contributed to the speech signal processing community since 1998. Although SPTK has reached over a hundred thousand downloads, the concepts as well as the features have not yet been widely disseminated. This paper gives an overview of SPTK and demonstrations to provide a better understanding of the toolkit. We have recently developed its differentiable PyTorch version, *diffsptk*, to adapt to advancements in the deep learning field. The details of diffsptk are also presented in this paper. We hope that the toolkit will help developers and researchers working in the field of speech signal processing.

**Index Terms**: digital signal processing, open-source software, differentiable DSP

## 1. Introduction

There are many applications using speech signals such as text-to-speech synthesis, singing voice synthesis, speech recognition, speaker recognition, and speech coding. To further the research and development of speech products, it would be beneficial to develop an open-source, general-purpose speech signal processing toolkit.

The Speech Signal Processing ToolKit (SPTK) was originally developed and used in the research group of Satoshi Imai and Takao Kobayashi at Tokyo Institute of Technology in 1990s. The tools can be used via a command-line interface (CLI) on a UNIX environment. Some of the tools were repackaged by Keiichi Tokuda as the organizer in collaboration with Takashi Masuko and Kazuhito Koishida, and then distributed as SPTK version 1.0[1] in 1998. The source code of the distribution including data processing, graph drawing, sample rate conversion, Fourier transform, speech analysis, speech synthesis, and vector quantization was written in the traditional C language. In 2000, SPTK version 2.0[2] was released with an additional 30 tools, bringing the total to about 100 tools. Note that versions 1.0 and 2.0 were not approved for commercial use. Then SPTK version 3.0[3] was distributed in 2002 with the modified BSD license to be more suitable for product development. The only difference from version 2.0 was the license. SPTK version 3.0 was then improved and maintained for the next several years, and matured into version 3.11 in 2017. These versions have been openly maintained on the SourceForge platform and have been widely used in the speech signal processing community [1, 2, 3, 4]. However, the source code was somewhat unreadable, which made it difficult to understand the implemented algorithms and modify the source code. In addition, the implemented features were not sufficiently portable because the source code was written in C.

To address these issues, we rewrote SPTK version 3.11 in C++ while retaining its core design, which will be described in detail in the next section. Although it can be rewritten in Python, we selected C++ for processing speed and compatibility with embedded platforms. We only used Python for graph drawing to generate modern images using sophisticated Python plotting libraries. The new SPTK was released as version 4.0[4] in 2021 with additional tools and continues to be maintained in the public GitHub repository. With the migration, the license was changed from the modified BSD license to the Apache License 2.0.

The new version of SPTK is readable and highly portable; however, the implemented features are not compatible with modern deep learning frameworks. The integration of deep learning with speech signal processing techniques is a research area of growing interest, and its effectiveness has begun to show in various contexts [5, 6, 7, 8, 9, 10]. Thus, we started to export the SPTK features to be compatible with one of the most widely used deep learning frameworks, PyTorch [11]. The exported SPTK library has been publicly distributed as *diffsptk*[5] since 2022. It includes some special signal processing modules such as mel-cepstral analysis [12] and mel-cepstral synthesis filtering [13], which are not implemented in other signal processing libraries [10, 14]. Further details will be described in Section 4.

The remaining drawback of SPTK is that the concepts and features are not well explained, making it difficult for users to approach. To solve this problem, in this paper, we introduce the concepts of SPTK as an open-source speech signal processing library and present the main features of SPTK with demonstrations to help users understand the tools. We also present the concepts of the differentiable version of SPTK for differentiable digital signal processing.

### 1.1. Related work

Table 1 shows a summary of signal processing libraries. Although there is some overlap in the table, SPTK also offers unique features, particularly for speech analysis and synthesis. SPTK can be used as a complement to other libraries.

---

[1]https://sourceforge.net/projects/sp-tk/files/SPTK/SPTK-1.0/
[2]https://sourceforge.net/projects/sp-tk/files/SPTK/SPTK-2.0/
[3]https://sourceforge.net/projects/sp-tk/files/SPTK/SPTK-3.0/

[4]https://github.com/sp-nitech/SPTK/releases
[5]https://github.com/sp-nitech/diffsptk

## 2. Design

We design SPTK on the basis of the following policies:

- **Raw data format**: The data used in SPTK do not have any headers or structures. No data compression is used. The raw data format enables users to read the contents of data files immediately via a binary file dump. This is very helpful for checking the sanity of data in experiments. In addition, the data generated by SPTK can be used in other software through simple binary reading. This policy is opposite to other well-known libraries such as the Kaldi archive format (.ark), hierarchical data format (.hdf), and binary data format in NumPy (.npy). The data type used in SPTK is little-endian 64-bit double (version 4.0 or higher) or 32-bit float (version 3.11 or lower) in principle.

- **Standard I/O-based**: SPTK consists of over 100 commands. Most of the commands receive input data from the standard input and send the processed data to the standard output. This means that users can perform complex data processing by combining the SPTK commands using the pipe command (|) in Unix-like computer operating systems. The SPTK commands can chain with basic UNIX commands such as *cat*, *less*, and *wc*. This policy is unique to SPTK [16, 17] and makes it intuitive and easy to use. To prevent data contamination, error or warning messages from the SPTK commands are output to the standard error rather than the standard output.

- **Non-interactive**: The SPTK commands do not require interactive user inputs. The parameters that control data processing, e.g., frame shift in speech analysis, must be set via command line options beforehand.

- **Minimum requirements**: SPTK intentionally avoids the use of external libraries such as Eigen [18]. While importing more external libraries facilitates the development of SPTK, some users or systems may not be able to install the libraries due to their machine environments. Furthermore, using multiple libraries makes licensing complicated and less user-friendly. To avoid these problems, we have implemented signal processing algorithms from scratch, including the fast Fourier transform (FFT).

- **Thread-safe** (version 4.0 or higher): SPTK ensures thread safety for parallel data processing. A general C++ class in SPTK has a *Run* function to perform data processing. The *Run* function typically requires the reference of input data, the pointer of output data, and the pointer of buffers as the arguments. By using different buffers in different threads, users can perform data processing in parallel without unintended data access.

- **No memory leaks** (version 4.0 or higher): The older versions of SPTK have a risk of memory leaks. To avoid this, we use std::vector in the C++ standard template library instead of the malloc function for dynamic memory allocation. In addition, a memory mismanagement detector is used to check for memory leaks in testing.

## 3. Features

The section describes the main features in the current version of SPTK.

### 3.1. Data type conversion

One of the most frequently used SPTK commands is *x2x*. The command converts the input data type to a specific data type. In the following example, all values in the short-type example file (*data.short*) are increased by two times.

```
$ x2x +sd data.short | sopr -m 2 |
    x2x +da | less
```

The first *x2x* converts the short type to double type to process the example data in the other SPTK commands. The last *x2x* converts the double type to ASCII to show the processed example data on the screen.

### 3.2. Data rearrangement

The order of data in SPTK is represented by the following vector:

$$\begin{bmatrix} \boldsymbol{x}_0^\mathsf{T} & \boldsymbol{x}_1^\mathsf{T} & \cdots & \boldsymbol{x}_{N-1}^\mathsf{T} \end{bmatrix}^\mathsf{T}, \tag{1}$$

where $\boldsymbol{x}_n$ is a $D$-dimensional vector $[x_{n,1}, x_{n,2}, \ldots, x_{n,D}]^\mathsf{T}$ and $N$ is the length of data sequence. SPTK can rearrange the data on a CLI, e.g.,

$$\begin{bmatrix} \boldsymbol{x}_S^\mathsf{T} & \boldsymbol{x}_{S+1}^\mathsf{T} & \cdots & \boldsymbol{x}_E^\mathsf{T} \end{bmatrix}^\mathsf{T} \tag{2}$$

is obtained by *bcut* where $S \geq 0$ and $E < N$. In the example below, the 2nd and 3rd samples of the example data are extracted:

```
$ bcut -l 1 -s 2 -e 3 +s data.short
```

where -l 1 means $D = 1$ and +s assumes short-type input data. Slicing (*bcp*), concatenation (*merge*), reversing (*reverse*), delaying (*delay*), transpose (*transpose*) operations are also provided. More information can be found in the reference manual: https://sp-nitech.github.io/sptk/latest/.

### 3.3. Graph drawing

To better understand data visually, the SPTK commands make it possible to draw data (*fdrw*), waveforms (*gwave*), discrete signals (*gseries*), log-spectrum (*glogsp*), running log-spectra (*grlogsp*), spectrogram (*gspecgram*), and pole-zero (*gpolezero*). These commands are implemented in Python using the Plotly graphing library [19]. The examples in this paper use these commands but some options are omitted due to space limitations.

Table 1: *Summary of open-source signal processing libraries*

|  | SPTK3 | SPTK4 | diffsptk | SciPy [15] | TorchAudio [14] |
|---|---|---|---|---|---|
| Language | C | C++ | Python | Python | Python |
| UNIX-like commands | ✓ | ✓ | | | |
| Deep learning | | | ✓ | | ✓ |
| Community | SourceForge | GitHub | GitHub | GitHub | GitHub |
| License | BSD 3-Clause | Apache 2.0 | Apache 2.0 | BSD 3-Clause | BSD 2-Clause |

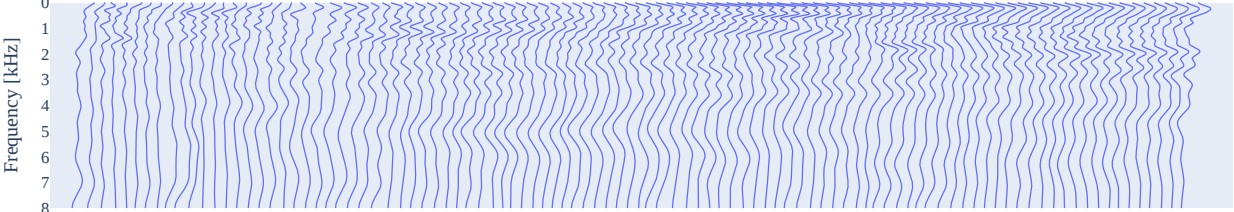

Figure 1: *Running spectra of the first 100 frames of the example data.*

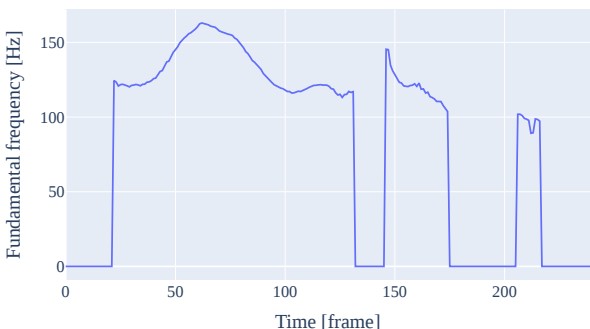

Figure 2: *A pitch contour of the example data.*

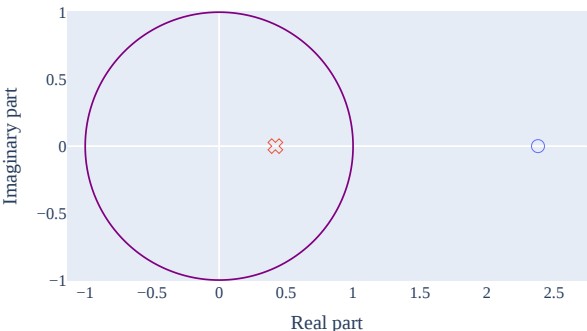

Figure 3: *Pole and zero of a first-order all-pass filter. Small circle and x-mark represent zero and pole, respectively.*

### 3.4. Spectral analysis

The example below computes spectra from a speech waveform using the short-term Fourier transform (STFT) with a 25-ms Blackman window and 5-ms frame shift:

```
$ x2x +sd data.short |
    frame -p 80 -l 400 |
    window -l 400 -L 512 |
    spec -l 512 > data.sp
```

where the FFT length is set to 512. Acoustic features can then be extracted from the spectra via autocorrelation analysis (*acorr, lpc*), adaptive mel-cepstral analysis (*amgcep*) [20, 21], or mel-generalized cepstral analysis (*mgcep*) [12, 22, 23]. In the following example, 24-th order mel-cepstral coefficients are extracted from the obtained spectra.

```
$ mgcep -m 24 -l 512 -a 0.42 -q 0 \
    < data.sp > data.mc
```

Figure 1 shows the running spectra computed from the mel-cepstral coefficients of the first 100 frames of the example data. The figure is generated by the following command.

```
$ mgc2sp -m 24 -l 512 -a 0.42 data.mc |
    grlogsp -l 512 -t -x 16 -e 99 \
            -H 500 -W 2000 spec.pdf
```

### 3.5. Pitch analysis

The extraction of pitch contours of speech is an important procedure in signal processing. SPTK provides a wrapper of sophisticated pitch extraction algorithms independently developed by third parties. The current implemented algorithms are RAPT [24], SWIPE' [25], REAPER [26], and DIO [27]. In the following example, a pitch contour of the example data is extracted by the RAPT algorithm with a 5-ms frame-shift.

```
$ x2x +sd data.short |
    pitch -a 0 -s 16 -p 80 > data.pit
```

Figure 2 shows the extracted $F_0$ contour obtained by the following command.

```
$ sopr -magic 0 -INV -m 16000 -MAGIC 0 \
    < data.pit | fdrw -g f0.pdf
```

In the command, *sopr* converts pitch [sec] to $F_0$ [Hz]. Note that SPTK outputs an unvoiced symbol as 0 as a magic number. If $\log F_0$ is selected as the output format of pitch, the unvoiced symbol is represented as $-1e+10$. SPTK also provides a command used for pitch mark (GCI) extraction (*pitch_mark*).

### 3.6. Speech synthesis (linear time-variant filtering)

SPTK can reconstruct waveform from acoustic features given an excitation signal using a linear synthesis filter. The implemented synthesis filters are an all-zero digital filter using impulse response (*zerodf*), all-pole digital filter using linear predictive coding (LPC) coefficients (*poledf*) [28], all-pole lattice digital filter using PARCOR coefficients (*ltcdf*), line spectral pairs (LSP) digital filter using LSP coefficients (*lspdf*) [29], and mel-log spectrum approximation (MLSA) digital filter using mel-cepstral coefficients (*mglsadf*) [20, 30]. In the following example, the speech waveform is reconstructed from a simple excitation signal using the MLSA filter with the extracted mel-cepstral coefficients.

```
$ excite -p 80 data.pit |
    mglsadf -p 80 -m 24 -a 0.42 -P 7 \
    < data.mc | x2x +ds -r > syn.raw
```

The commands for checking the stability of these synthesis filters are provided (*lpccheck, lspcheck, mlsacheck*).

### 3.7. Linear time-invariant filtering

A signal can be processed using a finite/infinite impulse response (FIR/IIR) digital filter. The example below shows how to apply a first-order all-pass filter to the example data:

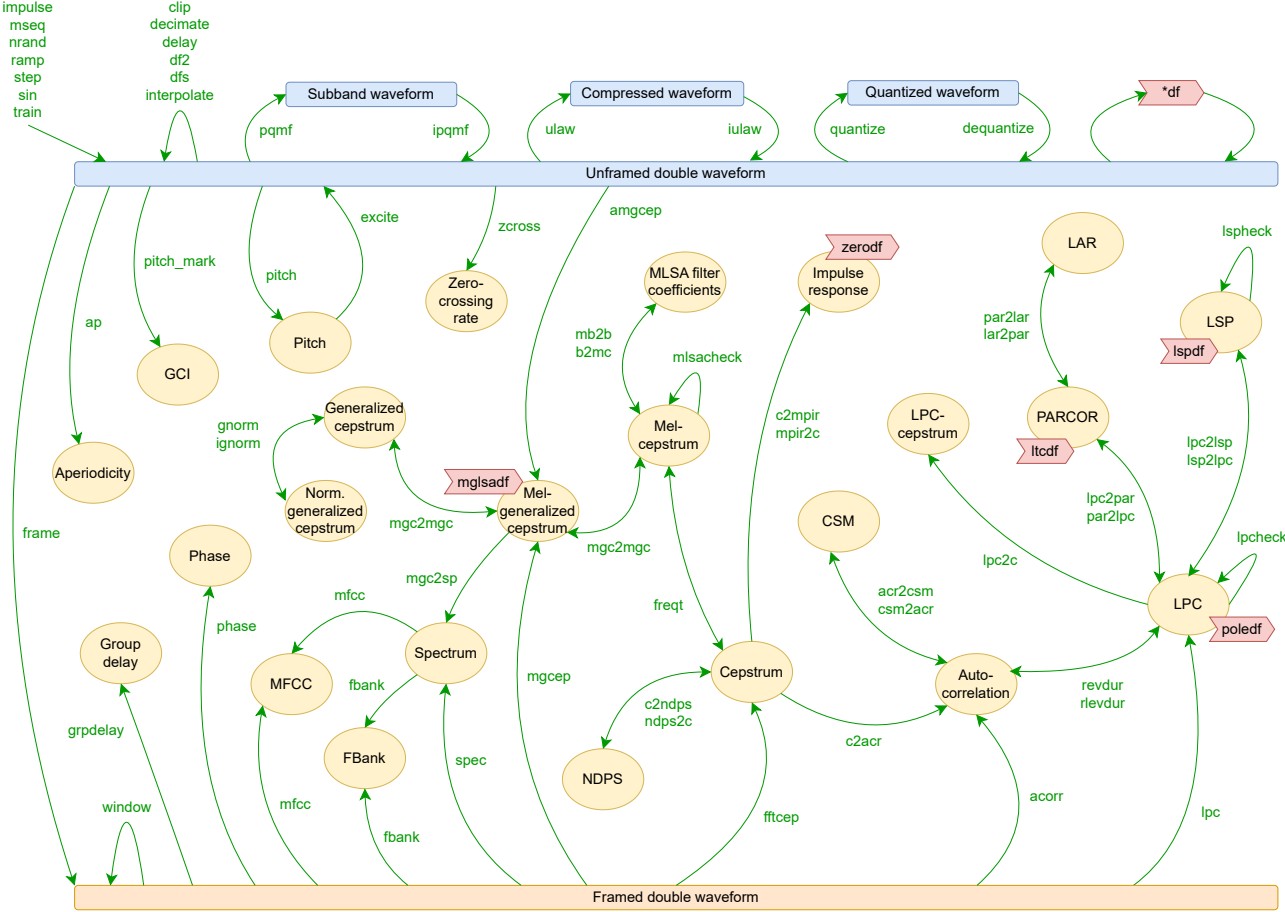

Figure 4: *Relationship between SPTK commands (green strings) and data representations (yellow circles). This block diagram is inspired by Mitch Bradley's design, who kindly provided us the block diagram for previous versions of SPTK.*

```
$ x2x +sd data.short |
    dfs -a 1 -0.42 -b -0.42 1 > data.out
```

where the transfer function is

$$H(z) = \frac{-0.42 + z^{-1}}{1 - 0.42z^{-1}}. \quad (3)$$

The poles and zeros of a digital filter can be computed and plotted in the $z$-plane. Figure 3 plots the pole and zero of the first-order all-pass filter $H(z)$. The figure is generated as follows.

```
$ echo 1 -0.42 | x2x +ad |
    root_pol -m 1 -q 0 > data.p
$ echo -0.42 1 | x2x +ad |
    root_pol -m 1 -q 0 > data.z
$ gpolezero -p data.p -z data.z pz.pdf
```

### 3.8. Mel-filter bank analysis

For speech recognition or speaker recognition, SPTK supports mel-filter bank analysis (*fbank, mfcc*) based on HTK [31]. In the following example, 12-order mel-frequency cepstrum coefficients (MFCCs) and the energy with their first derivatives are extracted. A time-invariant pre-emphasis filter is applied to the example data before the extraction.

```
$ x2x +sd data.short |
    frame -l 400 -p 160 -n 1 |
```

```
dfs -b 1 -0.97 |
window -l 400 -L 512 -w 1 -n 0 |
mfcc -l 512 -n 40 -c 22 -m 12 \
    -L 64 -H 4000 -s 16 -q 4 -o 1 |
delta -m 12 -d -0.5 0 0.5 > data.mfc
```

### 3.9. Parameter transformation

The relationship between the SPTK commands and data representation is illustrated in Fig. 4. As the figure shows, spectral parameters can be reversibly transformed into other representations including impulse response, autocorrelation, LPC coefficients, PARCOR coefficients, LSP coefficients [32], log area ratio (LAR), composite sinusoidal modeling (CSM) [33], cepstral coefficients, mel-cepstral coefficients, negative derivative of phase spectrum (NDPS) [34], and more.

### 3.10. Vector quantization

For speech coding, a codebook for vector quantization can be generated using the Linde-Buzo-Gray algorithm [35]. The codebook is obtained by gradually increasing the codebook size. Vector quantization can then be performed with the generated codebook. The example below computes a codebook of the extracted mel-cepstral coefficients and reconstructs them from the encoded vector indices.

```
$ lbg -m 24 -e 32 data.mc > mc.cb
$ msvq -m 24 -s mc.cb < data.mc |
    imsvq -m 24 -s mc.cb > data.mc.dec
```

Multi-stage (redidual) vector quantization can be performed by stacking −s option.

### 3.11. Subband decomposition

Subband analysis and synthesis using pseudo-quadrature mirror filters (PQMFs) [36, 37] is supported in SPTK. The filter coefficients will be designed to have the desired stopband attenuation. The example below decomposes the example data to two-channel signals and reconstructs them from the decomposed signals.

```
$ x2x +sd data.short |
    pqmf -k 2 -m 20 |
    decimate -l 2 -p 2 |
    interpolate -l 2 -p 2 |
    sopr -m 2 |
    ipqmf -k 2 -m 20 |
    x2x +ds -r > syn.raw
```

### 3.12. Voice conversion

SPTK also provides the commands for Gaussian mixture model (GMM)-based voice conversion [38, 39]. The alignment between the feature vector sequence of a source speaker and a target speaker can be obtained by dynamic time warping (*dtw*). The joint feature vector consisting of the feature vectors of the source and target speakers can then be modeled by GMMs (*gmm*). Finally, a feature vector sequence of the target speaker can be predicted from the trained GMMs and a given feature vector sequence of the source speaker (*vc*). Dynamic features can be easily appended to the feature vector sequences (*delta*) so that the smoothed feature vector sequence of a target speaker can be obtained.

### 3.13. Distance calculation

It is important to evaluate experimental results in terms of objective metrics. SPTK can compute signal-to-noise ratio (SNR), root-mean-square error (RMSE), and cepstral distance [40] for the metrics. These commands accept two inputs as follows.

```
$ cdist -m 24 -o 0 data.mc data.mc.dec |
    x2x +da
```

This is an example of mel-cepstral distance computed in decibels between the original and reconstructed data and shown on the screen.

### 3.14. Statistics calculation

SPTK can be used to easily compute statistics of data by using a single command, e.g, average (*average*), summation (*vsum*), mean, covariance (*vstats*), median (*median*), minimum, and maximum (*minmax*). The example below shows the mean vector of the extracted mel-cepstral coefficients on the screen.

```
$ vstat -m 24 -o 0 data.mc | x2x +da
```

## 4. PyTorch version

Incorporating digital signal processing techniques with deep learning is an area of growing interest. Although signal processing libraries for deep learning such as TorchAudio [14] have already been distributed, they have not implemented the core features of SPTK. Thus, we have re-implemented most of the SPTK features on the basis of a deep learning framework and provided them as a supplemental differentiable digital signal processing library. The library is named *diffsptk* as it is a differentiable version of SPTK. We selected PyTorch [11] as a deep learning framework because it is widely used by the deep learning community and easy to use. The license of diffsptk is the Apache License 2.0, which is the same as that of SPTK.

We design diffsptk on the basis of the following policies:

- **Non-recursive**: SPTK originally written in C/C++ involves recursive algorithms in the implementation within frequency warping [41], parameter transformations [23, 42], digital filtering [30], etc. This is suitable for non-parallel computation but not for deep learning using GPU parallel computing. To avoid slow training/inference, we have replaced the recursive implementation with a non-recursive one using mathematical techniques such as matrix multiplication and the FFT.
- **Dimension-last**: A neural network module in PyTorch accepts *tensors* as input and output. In diffsptk, the shape of the tensors is basically assumed as $(B, N, D)$ rather than $(B, D, N)$, where $B$ is the mini-batch size, $N$ is the data length, and $D$ is the data dimensions. This is more intuitive because the shape $(B, N, D)$ is compatible with the C version of SPTK described in Eq. (1).
- **Precomputed**: The parameters corresponding to the command line options in the SPTK commands must be set via the constructor of a PyTorch module, not the forward function used at runtime. This is consistent with the C++ class in SPTK. The policy enables us to reduce computation time at runtime by performing calculations in advance that depend only on the parameters and not input data.

### 4.1. Spectral analysis

The following Python code emulates the example which extracts the mel-cepstral coefficients from the example data as described in Subsection 3.4.

```python
import diffsptk

# Read the example data.
x, sr = diffsptk.read(
    "data.short",
    format="RAW",
    samplerate=16000,
    channels=1,
    subtype="PCM_16"
)

# Prepare PyTorch modules.
frame = Frame(400, 80)
window = Window(400, 512)
spec = Spectrum(512)

# Compute power spectrum.
sp = spec(window(frame(x)))

# Prepare a mel-cepstral analyzer.
mgcep = diffsptk.MelCepstralAnalysis(
    24, 512, 0.42, n_iter=30
)

# Extract mel-cepstal coefficients.
mc = mgcep(sp)
```

Using diffsptk is intuitive and compatible with the SPTK commands, as shown by the above code.

### 4.2. Pitch analysis

As with SPTK, pitch extraction relies on third-party libraries. The current implemented algorithm based on neural networks is CREPE [43]. In the following example, a pitch contour of the example data is extracted as in Subsection 3.5.

```
# Read data as in the previous example.
pit = diffsptk.Pitch(
    frame_period=80,
    sample_rate=sr,
    algorithm="crepe"
    f_min=80,
    f_max=180,
    out_format="pitch",
)(x)
```

The pitch embedding can be obtained instead of pitch by changing the out_format option.

### 4.3. Speech synthesis

We have re-implemented the synthesis filters on the basis of the FIR filter by approximating the IIR filter [13] to make the filters GPU-friendly. The example corresponding to Subsection 3.6 is as follows.

```
# Generate an excitation signal.
excite = diffsptk.ExcitationGeneration(
    frame_period=80
)
e = excite(pit)

# Synthesize waveform.
mlsa = diffsptk.MLSA(
    24,
    frame_period=80,
    alpha=0.24,
    taylor_order=30
)
y = mlsa(e, mc)

# Write reconstructed waveform.
diffsptk.write(
    "syn.raw",
    y,
    sr,
    format="RAW",
    subtype="PCM_16"
)
```

For more details, please see the reference manual: https://sp-nitech.github.io/diffsptk/latest/.

## 5. Conclusions

We have presented an overview of the core design, features, history, and current progress of SPTK. SPTK provides useful UNIX-like signal processing commands and a library to support product developments and research experiments. The differential version of SPTK has begun to be distributed to adapt to the deep learning paradigm.

## 6. Acknowledgements

We thank all of the contributors to SPTK. The names of the main contributors are listed at https://github.com/sp-nitech/SPTK/blob/master/README.md.

This work was partly supported by JSPS KAKENHI Grant Number JP22H03614.

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
