# OpenReview forum: "SPTK4: An Open-Source Software Toolkit for Speech Signal Processing"
_Interspeech.org/2023/Workshop/SSW — SSW12_

### Official Review · Reviewer_7ANm · 2023-06-04
**The paper presents an overview of the signal processing open source tool SPTK , as well as introduces (briefly) the new differentiable version of the tool, diffsptk**

**Rating:** 7
**Confidence:** 4

**Review:**

SPTK toolbox is known in the speech synthesis community since parametric speech synthesis was introduced. It has several unique features, including an easy unix-like command line interface. However, it is less known to the new generation of speech synthesis researchers, that are familiar with NN-based TTS only.
The authors of this paper present the capabilities of the SPTK interface and introduce the new differentiable version of this tool, diffsptk, based on pytorch. The paper is useful and interesting, however, it would be beneficial for the community to add more details on the new differentiable tool, including some Pythonic examples of its usage.

minor comments on the paper:
1. in formula (1) T, which indicates the length of the sequence, coincides with T, indicating the transpose operation. Please use another letter

---

### Official Review · Reviewer_23vd · 2023-06-04
**Description of SPTK adding new developments**

**Rating:** 7
**Confidence:** 5

**Review:**

quality:
The paper describes the development of SPTK including some new developments like diffsptk. The paper is generally well written and readable.

clarity:
The description of the piped commands is sometimes not so useful as not all commands are explained in the text. Here it would have been more useful to have less commands with detailed description. Also the data rearrangement 3.2 is unnecessarily complicated.

originality:
Some of the things are already part of the SPTK documentation, I would have wished more information on diffsptk.

significance:
Is is interesting to read about the development of SPTK, it's design choices, and how it integrates with new developments in deep learning.

---

### Decision · Program_Chairs · 2023-06-14

**Decision:**

Accept

**Comment:**

SSW2003 received 45 papers. The acceptance rate is 82%. We are pleased to inform you that your paper has been accepted by the SSW2023 Program Committee. Please read the reviews carefully and submit your camera-ready paper by June 28th. Most reviewers performed a detailed review. Please answer to their questions and consider their comments. Note that camera-ready papers are credited with one extra page to allow authors to consider reviewers’ suggestions. So max 7 pages in total including figures & refs.
The deadline for submitting the revised version (with full non-anonymized authors and refs!) is 28th June.